# 3D-RelNet: Joint Object and Relational Network for 3D Prediction

## Abstract

We propose an approach to predict the 3D shape and pose for the objects present in a scene. Existing learning based methods that pursue this goal make independent predictions per object, and do not leverage the relationships amongst them. We argue that reasoning about these relationships is crucial, and present an approach to incorporate these in a 3D prediction framework. In addition to independent per-object predictions, we predict pairwise relations in the form of relative 3D pose, and demonstrate that these can be easily incorporated to improve object level estimates. We report performance across different datasets (SUNCG, NYUv2), and show that our approach significantly improves over independent prediction approaches while also outperforming alternate implicit reasoning methods.

## 1 Introduction

A single 2D image can induce a rich 3D perception. When we look at an image, we can reason about its 3D layout, the objects in the image, their shape, extent, relationships *etc.*. This is really surprising given that going from a 2D projection to a 3D model is inherently ill-posed. How are we able to solve this problem? Humans rely on regularities in the 3D world in order to do so – this helps us discard many improbable solutions in 3D and reason about more likely ones. This regularity exists at the scene level - indoor scenes have roughly perpendicular walls; object level - chairs have similar shapes; and in local object relationships - chairs are close to tables, monitors are on top of tables *etc.*. A decade ago, a lot of work in computer vision focused on using all the three levels of regularities. For example, a lot of work focused on object-centered 3D prediction (Fidler et al., 2012), scene-level 3D prediction (Hedau et al., 2009), and multi-object 3D reasoning (Jiang et al., 2012). However, in recent years, since the advent of ConvNets, a vast majority of computer vision approaches do not leverage these object-object relationships, and instead reason about each object independently.

In this paper, we attempt to take a holistic view of the 3D prediction problem and note that solving the 3D prediction problem would require incorporation of all the three cues. We believe there are three fundamental questions that need to be answered to design this holistic architecture: (a) What is the right representation for object level 3D prediction?; (b) How do we represent object-object relationships and how do we predict them from pixels?; (c) Finally, how to incorporate object-object relationships with object-level modules. This paper builds upon the recent success in (a) and investigates how to model relationships and incorporate them into our 3D prediction framework.

So, how do we model relationships and estimate them from pixels? There is a whole spectrum of possible approaches. On one end of the spectrum is a complete end-to-end approach. Some examples of these include Interaction Networks (Battaglia et al., 2016) or Graph Convolutional Networks (Kipf & Welling, 2016). Both these methods provide a mechanism for object features in the scene to effect each other, thereby allowing an implicit modeling of relationships among them. However, as we show in our experiments, these end-to-end approaches disregard the structural information which might be crucial for modeling the relationships. The other end of spectrum is to use category-based image-agnostic pairwise priors (Zhao & Zhu, 2011) to model relationships. A drawback is that these priors are often too strong to generalize and it is better to learn them (Jiang et al., 2012).

When it comes to the final question of how does one incorporate relationships to improve 3D prediction, the answer is even murkier. One classical approach is to use graphical models such as

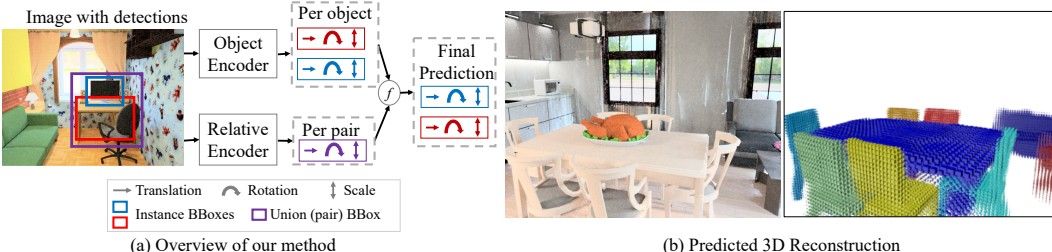

(a) Overview of our method        (b) Predicted 3D Reconstruction

Figure 1: **(a) Approach Overview:** We study the problem of layout estimation in 3D by reasoning about relationships between objects. Given an image and object detection boxes, we first predict the 3D pose (translation, rotation, scale) of each object and the relative pose between each pair of objects. We combine these predictions and ensure consistent relationships between objects to predict the final 3D pose of each object. **(b) Output:** An example result of our method that takes as input the 2D image and generates the 3D layout.

CRFs (Lafferty et al., 2001; Schwing et al., 2012). However, these classical approaches have usually provided little improvements over object-based approaches. Our key insight is to incorporate structural information in end-to-end systems. Specifically, we model and predict pairwise relationships in the translation, rotation and scale space. One advantage of using this structured relationship space is that the incorporation of relationships into object-level estimates is simple yet effective. But how do we predict these pairwise relationships from pixels? Our paper investigates several design choices and proposes a simple architecture. Our method demonstrates significant improvement in performance across multiple metrics and datasets. As we show in our experiments, this modeling of relationships in this structured space provides a huge **6 point AP** improvement in detection settings over current state-of-the-art 3D approaches. We will release our code for reproducibility.

## 2 RELATED WORK

Dating back to the first thesis written on computer vision (Roberts, 1963), inferring the underlying 3D structure of an image has been a long-standing goal in the field. While we have explored several 3D scene representations over the years, *e.g.*, depth (Saxena et al., 2009; Eigen et al., 2014), qualitative 3D (Hoiem et al., 2005; Gupta et al., 2010), manifolds (Osadchy et al., 2007) or volumetric 3D (Choy et al., 2016; Girdhar et al., 2016; Song et al., 2017) , the prevalent paradigm is still the one followed in Roberts' seminal work – that of inferring a 3D scene in terms of the shape and pose of the underlying objects.

The initial attempts under this paradigm (Guzmán, 1968; Huttenlocher & Ullman, 1990) focused on placing known object instances to match image evidence, relying on matching edges, corners *etc.* to fit the known shape templates to images. Subsequent approaches have focused on a more general setting of reconstructing scenes comprising of novel objects, and leverage either explicit or implicitly learned category level priors for pose and shape estimation, typically relying on a deformable shape space (Cashman & Fitzgibbon, 2013; Kar et al., 2015) or template CAD models (Lim et al., 2013; Aubry et al., 2014; Li et al., 2015; Bansal et al., 2016; Izadinia et al., 2017) for the latter inference. Current CNN based incarnations of these approaches, driven by the abundant success of deep learning and availability of annotated data, have further improved the results for pose estimation (Tulsiani & Malik, 2015; Pavlakos et al., 2017), and have also been extended to joint shape and pose inference of the objects present in a scene (Kundu et al., 2018; Tulsiani et al., 2018).

A common characteristic amongst these approaches is the reasoning at a per-object level. While the object-centric nature is certainly desirable as a representation, we argue that reasoning independently for each object to infer this representation is not, as it does not allow leveraging the relationships between the entities in a scene. We propose a method that also uses object-centric representations but goes beyond independent reasoning per object.

We are of course not the first to pursue reasoning about relationships between entities in a scene. Several previous approaches focus on the goal of predicting various relations e.g. human-object interactions (Gupta & Malik, 2015; Gkioxari et al., 2018), object-object interactions (Gupta et al., 2010; Lu et al., 2016), object-scene interactions (Lee et al., 2010) *etc.* While these works pursue

**Figure 2: Approach Details:** We use the instance encoder to create an embedding for each object (instance) in the scene. The instance decoder uses this embedding to predict a pose for each object independently. The relative encoder takes each pair of instances and their embeddings to output an embedding for the pair. The relative decoder predicts a relationship (relative pose) between the pairs. We combine these relative and per-object predictions to predict final pose estimates for each object in an end-to-end differentiable framework.

relation inference as the end goal, we instead aim to leverage these for a per-instance prediction task. In the context of incorporating relations for such per-instance prediction, there are two alternate ideologies. On the one hand, approaches pursuing 3D scene inference or generation (Zhao & Zhu, 2011; Lin et al., 2013; Choi et al., 2013; Fisher et al., 2012; 2011; Huang et al., 2018; Jiang et al., 2018) typically incorporate pairwise (or higher order) relations via explicit class-based priors regarding possible configurations and optimize predictions to adhere to these. This approach of explicitly modeling relations as a prior imposes the same constraints across all scenes, independent of the structure in the image, and is therefore not flexible enough and has difficulties in scaling up to arbitrary relations across arbitrary objects.

The alternate ideology for incorporating relationships is to eschew any explicit structure for these relations, and instead implicitly capture these via architectural changes to the CNNs, thereby allowing the features of objects to influence one another (Battaglia et al., 2016; Kipf & Welling, 2016). While this a generally applicable mechanism, it does not leverage several aspects regarding the structure of the problem – for 3D inference, specific relations like relative position, orientation are very relevant, and can be used in specific ways to influence per-instance predictions. Our approach leverages some aspects of both these ideologies – unlike the classical prior based approaches, we learn and infer these relations in a image-dependent context via a CNN, and unlike the purely implicit methods, we are more explicit about the structure and meaning of these relations.

## 3 APPROACH

Our goal is to predict the 3D pose and shape for all the objects in a scene. We observe that in addition to the visual cues per object, reasoning about relationships across them can further help our predictions, in particular for the 3D pose – a chair would be in front of a table, and of a compatible relative size, and therefore even if we are uncertain about the pose of one of these objects, *e.g.*, due to occlusion, these relationships can enable us to make accurate predictions.

We operationalize this insight in our method (see figure 1) that leverages both - independent per-object predictions alongwith predictions regarding relationships between them. We infer the final estimates for all the objects in the scene by integrating these two. We first formally describe the object-centric representations pursued and briefly review a recent per-instance prediction approach in section 3.1. We then introduce the relative representations in section 3.2 and present our network architecture that enables predicting these in section 3.3. In section 3.4 we discuss how these relative predictions are combined with the independent per-object predictions to yield the final 3D estimates for the objects. We show that optimizing the combination of these estimates (*i.e.*, final estimate) in a differentiable end-to-end framework helps improve the final per-object predictions.

### 3.1 INSTANCE SPECIFIC REPRESENTATIONS AND INFERENCE

We output the 3D pose of an object by predicting its shape in a canonical frame, and its scale, translation and rotation in camera frame. The shape is parametrized as a $32^3$ volumetric occupancy grid in a canonical space where the objects are upright, front-facing, normalized to a unit cube. The translation $\mathbf{t} \in \mathbb{R}^3$ and (logarithm of) anisotropic scale $\mathbf{s} \in \mathbb{R}^3$, and normalized quaternion $\mathbf{q}$

indicate the position, size and orientation of the object respectively. Following prior work (Tulsiani et al., 2018), we parametrize the rotation prediction as a classification task among fixed bins.

We build upon the recent work by (Tulsiani et al., 2018): they pursue a similar per-object representation, but make independent predictions across objects. We use their approach to obtain the independent predictions for each object. We briefly review their prediction framework but refer the reader to the paper for details about the representation and architecture. Their method uses an architecture similar to Fast R-CNN (Girshick, 2015), where the input image is encoded via convolutional layers, and for each object bounding box, an RoI pooling layer crops the corresponding features. These per-object features, in conjunction with coarse image feature and an encoding of the bounding box coordinates, are encoded to an instance-specific bottleneck representation from which the corresponding shape and pose are predicted.

We adopt a similar architecture, illustrated in figure 2, with the introduction of spatial coordinates as additional feature channels (see section 3.3) to obtain per-object (unary) predictions. We note that these predictions are obtained independently across objects, and do not incorporate reasoning across them. However, unlike previous work, these are subsequently integrated with relative predictions (section 3.4) to obtain final estimates.

## 3.2 RELATIONS AS RELATIVE REPRESENTATIONS

Given an image of a scene, we can infer that two chairs nearby might be of a similar size, a laptop kept above a desk, and a television facing the bed *etc.*. Thus, relative pose between objects captures an important aspect of their relationship in the scene.

Concretely, given two objects, say $A$ and $B$, we infer the relative pose from $A$ to $B$. The relative pose, akin to the absolute 3D pose, is factored into the relative translation, scale, and direction. The relative translation is defined as the difference of the absolute locations of the object in the camera frame $\mathbf{t}_{AB} = \mathbf{t}_B - \mathbf{t}_A$. Similarly, the relative scale is simply the ratio of the two object sizes, or equivalently a difference in logarithms $\mathbf{s}_{AB} = \mathbf{s}_B - \mathbf{s}_A$. Finally, we predict a relative *direction* $\hat{\mathbf{d}}_{AB}$, *i.e.*, the direction of object $B$ in the frame of object $A$: $\hat{\mathbf{d}}_{AB} \propto (\bar{R}_A) \, \mathbf{t}_{AB}$, where $\hat{\mathbf{d}}_{AB}$ is normalized to unit norm. We note that this parametrization, unlike relative rotation, helps us overcome some ambiguities due to symmetries, *e.g.*, the relative direction from a chair to a table in front of it is unambiguous, even if the table is symmetric.

## 3.3 NETWORK ARCHITECTURE

Our network has two branches – a per-object (unary) prediction module, and a relative prediction module. The architecture of both these branches is shown in figure 2. The former, as described in section 3.1, simply computes a per-object encoding and subsequently makes per-object predictions. The relative prediction module is then tasked with inferring the relative pose (section 3.2) for every pair of objects in the scene.

As depicted in figure 2, the relative prediction module takes as input: a) the encoding of both objects, and b) an encoding of the larger (union) bounding box containing both objects. Since this larger box may contain several additional objects beyond the ones of interest, we additionally give as input two channels with binary masks indicating the source and target bounding box extents respectively (denoted as 'object location image'). We also find it beneficial to concatenate the normalized per-pixel spatial coordinates as additional features, as it allows the network to easily reason about the absolute spatial location of the bounding box(es) under consideration. Finally, similar to the parametrization in section 3.1, we frame the relative direction prediction as a classification task among 24 bins, where the bins are computed by clustering relative directions across the training instances.

## 3.4 COMBINING PER-OBJECT AND RELATIVE PREDICTIONS

We saw in section 3.1 that we can obtain independent per-object predictions for the 3D shape and pose, and introduced the relative pose predictions and architecture in section 3.2 and section 3.3. We denote by $(\mathbf{t}_n, \mathbf{s}_n, \mathbf{q}_n)$ the per-object (unary) predictions for the translation, log-scale, and rotation respectively for the $n^{th}$ object, and similarly $(\mathbf{t}_{mn}, \mathbf{s}_{mn}, \mathbf{d}_{mn})$ denote the relative pose predictions

from the $m^{th}$ to the $n^{th}$ object. Using these, we show that we can obtain final per-instance predictions $(\mathbf{t}_n^*, \mathbf{s}_n^*, \mathbf{q}_n^*)$ that incorporate both, the unary and relative predictions.

**Translation and Scale Prediction.** The relative predictions give us linear constraints that the final predictions should ideally satisfy. As an example, we would want $\mathbf{t}_n^* - \mathbf{t}_m^* = \mathbf{t}_{mn}$. This can equivalently be expressed as $A_{mn}\mathbf{t}^* = \mathbf{t}_{mn}$, where $A_{mn}$ is a sparse row-matrix with the $m^{th}$ and $n^{th}$ entries as $(-1, 1)$, and $\mathbf{t}^*$ denoting the final translations for all the $N$ objects. We can similarly express all pairwise linear constraints as $A\mathbf{t}^* = \mathbf{t}_{\text{rel}}$, where $\mathbf{t}_{\text{rel}}$ denotes all the relative predictions, and $A$ is the appropriate sparse matrix.

In addition to satisfying these linear constraints, we would also like the final estimates to be close to the unary predictions. We can therefore incorporate both, the relative and the unary constraints via a system of linear equations, and solve these to obtain the final estimates.

$$\begin{bmatrix} \lambda\, I \\ A \end{bmatrix} \mathbf{t}^* = \begin{bmatrix} \lambda\, \mathbf{t} \\ \mathbf{t}_{\text{rel}} \end{bmatrix}; \quad \mathbf{t}^* = \begin{bmatrix} \lambda\, I \\ A \end{bmatrix}^+ \begin{bmatrix} \lambda\, \mathbf{t} \\ \mathbf{t}_{\text{rel}} \end{bmatrix} \tag{1}$$

Here $X^+$ denotes the Moore-Penrose inverse of matrix $X$, and $\lambda$ indicates the relative importance of the unary estimates. We can therefore obtain the final translation predictions $\mathbf{t}^*$ that integrate both, unary and relative estimates. We note that the final estimates are simply a linear function of the unary predictions $\mathbf{t}$ and the relative predictions $\mathbf{t}_{\text{rel}}$, and it is therefore straightforward to propagate learning signal from supervision for $\mathbf{t}^*$ to these predictions. While the description here focused predicting translation, as we represent scale using logarithm of the sizes, similar linear constraints apply. We can therefore similarly compute final predictions for the scales across objects $\mathbf{s}^*$.

**Rotation Prediction.** While the incorporation of unary and relative predictions can be expressed via linear constraints in the case of translation and scale, a similar closed form update does not apply for rotation because of the framing as a classification task and non-linearity of the manifold. Instead, we update the likelihood of the unary predictions based on how consistent each rotation bin is with the relative estimates.

We denote as $R^b$ the rotation matrix corresponding to the $b^{th}$ rotation bin, and use $\Delta(R, \mathbf{d}, \mathbf{t})$ to measure how inconsistent a predicted rotation $R$ is w.r.t the predicted relative direction distribution $\mathbf{d}$ and relative translation $\mathbf{t}$ (see appendix for details). Using these, we can compute the (unnormalized) negative-log likelihood distribution over possible rotations as follows:

$$\mathbf{q}_m^*(b) = \mathbf{q}_m(b) + \sum_n \Delta(R^b, \mathbf{d}_{mn}, \mathbf{t}_{mn}) \tag{2}$$

This update to compute $\mathbf{q}^*$ can equivalently be viewed as a single round of message passing, with the messages being of an explicit rather than an implicit form.

**Training Details.** We described above how the independent per-object predictions are analytically combined with the relative pose predictions to obtain the final estimates, and note that this integration process allows us to propagate learning signal from supervision on the final estimates back to the unary and relative predictions. Our training happens in two steps. In the first step, we train both unary and relative predictions independent of each other. We formulate the loss-function for each network similar to (Tulsiani et al., 2018). Specifically, we use regression losses for shape encoding, the (absolute and relative) translation and scale, and classification losses for the rotation and relative direction prediction. Note that as some objects might be rotationally symmetric, we allow multiple 'correct' bins for these and maximize the maximum probability across these. In the second step, after a few epochs, we train the whole model in joint manner. We add similar losses for the final pose predictions that are computed using both, unary and relative estimates. During inference, given the unary and relative predictions, we simply compute the final pose predictions via the optimization process described above. Additional details on optimization are provided in the appendix.

## 4 EXPERIMENTS

### 4.1 EXPERIMENTAL SETUP

**Datasets:** We use the SUNCG dataset (Song et al., 2017) which provides many diverse and detailed 3D models for houses. Following (Zhang et al., 2017; Tulsiani et al., 2018), we use the 2D renderings

of the houses and the corresponding parsed 3D house model information to get roughly 550k image and 3D supervision pairs. We follow the setup in (Tulsiani et al., 2018) and split this data into train (70%), val (10%) and test (20%); with objects classes - bed, chair, desk, sofa, table, and tv.

We also use the NYUv2 (Silberman et al., 2012) dataset which consists of 1449 real world images, and use the annotations by (Guo & Hoiem, 2013) to finetune the network trained on SUNCG using the same subset of object categories. This dataset has lower resolution images and serves well to check the generalization of our approach to real data. As the NYUv2 annotations use the same small set of 3D shapes across train and test images, we do not evaluate shape prediction.

**Metrics:** (Tulsiani et al., 2018) propose different metrics to measure the quality of various prediction components - detection, rotation, translation and scale. They also propose different thresholds $\delta$ for each of these components which are used to count a prediction as a true positive in the detection setting. We use these metrics and refer the reader to the appendix for a summary review.

**Baselines:** We use the following baseline methods:

- *Factored3D*: This method from (Tulsiani et al., 2018) reasons about each object independently and serves as a baseline to see how our relationship reasoning can improve performance.

- *GCN*: We use Graph Convolutional Networks (Kipf & Welling, 2016) to perform implicit relational reasoning. We use the Object Encoder (Figure 2) to obtain object embeddings $o_i$ for each object and then use a 2-layer GCN to obtain the final object embeddings for each object. These are then used to predict the 3D pose and shape for the objects.

- *InteractionNet*: We use the method from (Battaglia et al., 2016) as an alternate way to perform implicit reasoning over the object embeddings. We compute an 'effect' embedding $e_{AB}$ for each ordered tuple $(o_A, o_B)$ via a learned MLP, and update the object embeddings by aggregating these as $o_A + \max_B(e_{AB})$, and use these for per-object predictions.

Note that while the latter two baselines can implicitly reason about relationships, they ignore the underlying structure (how relative translations affect translation *etc.*).

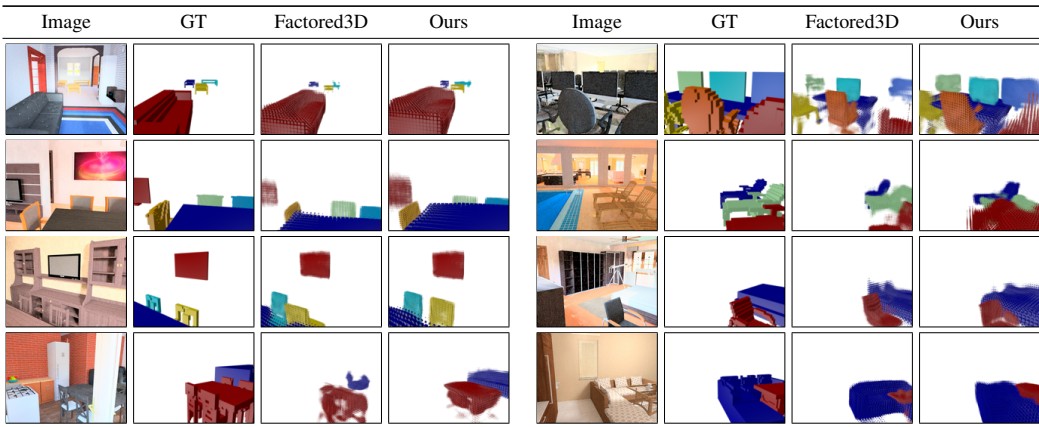

**Figure 3:** We visualize sample prediction results in the setting with known ground-truth boxes. Our method produces better estimates than the baseline, *e.g.*, bottom right where the distance between the chairs and table is predicted correctly, and the pose of the yellow chair is corrected. Best viewed in color.

## 4.2 USING GROUND TRUTH BOXES

We first analyze all methods in the setting where we are given the ground truth bounding boxes. In this setting, we can analyze just the 3D prediction quality without the additional variance introduced due to imperfect detection. During training, we train all the methods on ground-truth boxes as well as object proposals (obtained using (Zitnick & Dollár, 2014)) which have an IOU $\geq 0.7$ and at test time, we evaluate them only on ground truth boxes. We evaluate the final (combined) output of our method. As shown in Table 1, our method generates higher quality predictions across both translation and scale where it can outperform the baselines on the mean, median and % error. It is

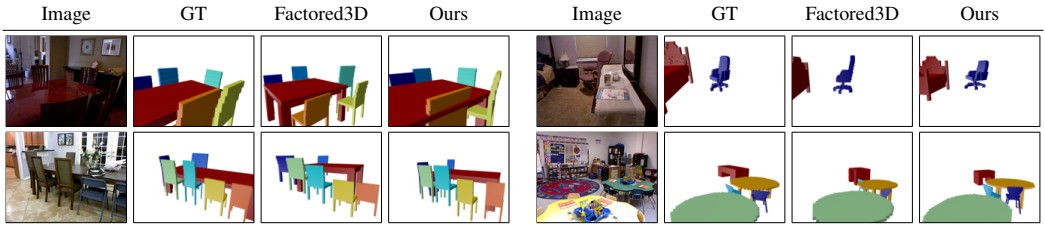

**Figure 4:** We visualize sample prediction results on the NYUv2 dataset in the setting with known ground-truth boxes. Our method produces better estimates than the baseline, *e.g.*, for the top left image we see that the chairs are better aligned by our method as compared to the baseline. We use the ground-truth meshes for visualization due to lack of variability in shape annotations for learning.

| Dataset | Method | Translation (meters) | | | Rotation (degrees) | | | Scale | | |
|---|---|---|---|---|---|---|---|---|---|---|
| | | Median | Mean | (Err ≤ 0.5m)% | Median | Mean | (Err ≤ 30°)% | Median | Mean | (Err ≤ 0.2)% |
| SUNCG | Factored3D | 0.28 | 0.39 | 79.5 | **4.56** | 19.91 | 86.4 | 0.16 | 0.25 | 58.4 |
| | InteractionNet | 0.28 | 0.37 | 80.0 | 4.58 | 20.19 | 86.4 | 0.11 | **0.19** | 68.6 |
| | GCN | 0.26 | 0.38 | 79.3 | 4.60 | 20.45 | 86.0 | 0.11 | 0.20 | 69.1 |
| | Ours | **0.23** | **0.33** | **84.0** | 4.58 | **19.82** | **86.6** | **0.10** | **0.19** | **69.7** |
| NYUv2 | Factored3D | 0.49 | 0.62 | 51.0 | 14.55 | 42.55 | 63.8 | 0.37 | 0.40 | 18.9 |
| | Interaction Net | 0.45 | 0.59 | 56.2 | **13.34** | **38.7** | **67.6** | 0.36 | 0.39 | 20.1 |
| | GCN | 0.45 | 0.60 | 55.6 | 14.22 | 41.63 | 65.7 | 0.37 | 0.40 | 19.2 |
| | Ours | **0.41** | **0.54** | **60.9** | 14.00 | 39.60 | 67.0 | **0.33** | **0.38** | **21.7** |

**Table 1:** We use **ground truth boxes** at test time and compare the 3D prediction performance of the methods. We show the median and mean error (lower is better) and the % of samples within a threshold (higher is better).

also worth noting that adding any form of relationships modeling, like in GCN or InteractionNet, gives a boost over doing per-object predictions like in Factored3D. Our structured reasoning and inference about object relationships provides a boost over other pairwise models such as GCN and InteractionNet. Our performance gain over the baseline holds across SUNCG and NYUv2 datasets demonstrating the generalization of our method.

**Qualitative Results:** In figure 3, we show a few results of our method and the baseline (Factored3D). We see that our method can correct many error modes (relative positions and poses of objects) compared to the baseline. We observe similar trends on the NYUv2 dataset (figure 4).

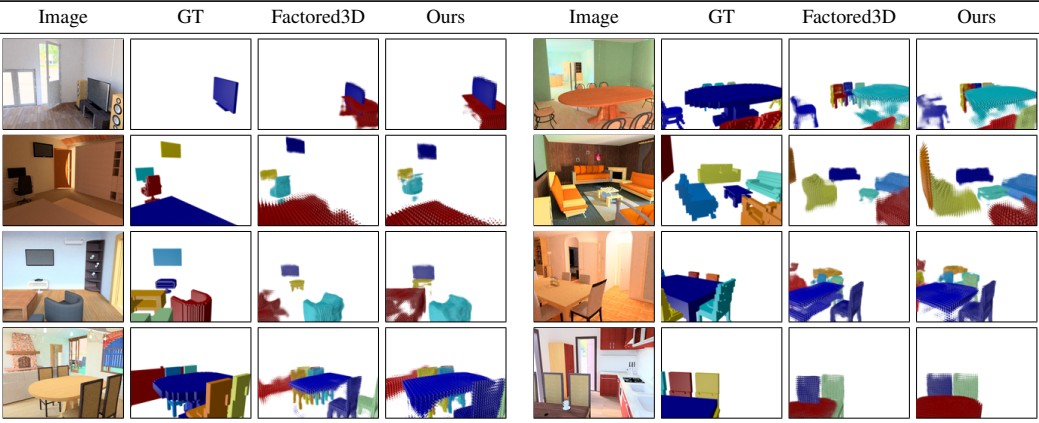

**Figure 5:** We visualize sample prediction results in the detection setting. Our method produces better estimates than the baseline, *e.g.*, in the first image the television set and the table are better placed. The colors only indicate separate instances, and do not correspond between the ground-truth and the predicted representations.

## 4.3 DETECTION SETTING

In this setting, we test the learned models using detections from a pre-trained object detector (taken from (Tulsiani et al., 2018)). Thus, we can now evaluate the robustness of these methods when the

| Method | all | box2d + trans | box2d + rot | box2d + scale |
|---|---|---|---|---|
| Factored3D | 21.72 | 49.28 | **62.77** | 33.56 |
| Interaction Net | 26.42 | 50.26 | 61.75 | 41.66 |
| GCN | 24.76 | 48.63 | 61.61 | 39.97 |
| Ours | **27.76** | **54.38** | 62.72 | **41.83** |

**Table 2:** We report the mean Average Precision (mAP) values for the **detection setting** for SUNCG. In each column, we vary the criteria used to determine a true positive. This helps us analyze the relative contribution of each component (translation, rotation, scale) to the final performance.

| Method | Translation (meters) | | | Rotation (degrees) | | | Scale | | |
|---|---|---|---|---|---|---|---|---|---|
| | Median | Mean | (Err $\leq$ 0.5m)% | Median | Mean | (Err $\leq 30°$)% | Median | Mean | %(Err $\leq$ 0.2)% |
| Multi-task (MT) | 0.25 | 0.35 | 81.9 | 4.55 | 19.33 | 86.7 | 0.11 | 0.20 | 68.2 |
| MT + combine test only | **0.23** | 0.34 | 83.5 | **4.51** | **19.30** | **86.9** | 0.11 | **0.19** | 68.4 |
| MT + combine train only | 0.25 | 0.36 | 82.0 | 4.63 | 20.02 | 86.1 | 0.11 | **0.19** | 68.9 |
| MT + combine train & test (Ours) | **0.23** | **0.33** | **84.0** | 4.58 | 19.82 | 86.6 | **0.10** | **0.19** | **69.7** |

**Table 3:** We study the effect of **combining the unary and relative predictions** at various stages. The multi-task model does not combine them, whereas the next two models combine them either at train or test time. Our method that jointly optimizes these predictions (and their combination) at both train and test time shows the biggest improvement.

input object boxes are not pristine. In Table 2, we see the mean Average Precision values for different criteria used to define true positives (see appendix for metrics). As an example, in the first column a true positive satisfies IOU (box2d) threshold with the ground truth, is close in scale, rotation, translation, shape thresholds. Each subsequent column examines one criteria so we can analyze their accuracy with detections. We see (in the first column) that our method provides a significant gain of **6 points** in mAP over the baseline. Relationship modeling (GCN and InteractionNet) provides benefit in the detection setting too. However, our structured reasoning outperforms these methods, with most of the gains coming from predicting higher quality scale and translation. We visualize some predictions showing the difference between our approach and the baseline in figure 5. Furthermore, in Table 4 we evaluate the our method on the NYUv2 in the detection setting, and we achieve similar trends in performance with respect to the baseline. Due to increased difficulty of the task on NYU we observe relative lower mAP scores.

## 4.4 EFFECT OF COMBINATION AND OPTIMIZATION

Combining the unary and relative predictions provides benefit at the training time because it allows the model to modify its unary and relative predictions so that they are more 'compatible' with each other. We quantify this in Table 3 using the SUNCG dataset. We report the performance of a purely multi-task version ('MT') that only predicts the unaries and relative pose values, without ever combining them. The 'MT + combine test only' combines these during inference (but not training). Finally, the 'MT + combine train only'. We also compare to a method that only uses this combination at train time, and finally report our full method as a reference. We see that combining at either train or test alone performs better than pure multi-task learning. This shows the importance of the unary and relative predictions interacting with each other. Combining these at both train and test time, and jointly optimizing like in our method performs the best.

| Method | all | box2d + trans | box2d + rot | box2d + scale |
|---|---|---|---|---|
| Factored3D | 5.30 | 17.17 | 20.36 | 28.36 |
| Interaction Net | 7.57 | 19.92 | 20.39 | 30.93 |
| GCN | 6.49 | 17.39 | 21.85 | 30.42 |
| Ours | **8.49** | **21.16** | **22.81** | **31.91** |

**Table 4:** We report the mean Average Precision (mAP) values for the **detection setting** for NYU. Note that the threshold for scale is different from the one used in SUNCG Table 2. Added after reviews

## 5 DISCUSSION AND FUTURE WORK

We proposed a method to incorporate relationship based reasoning in the form to relative pose estimates for the task of 3D scene inference. While this allowed us to significantly improve over existing approaches that reason independently across objects, numerous challenges still remain to be addressed. In our approach, we only leveraged pairwise relations among the objects in a scene, and it would be interesting to pursue incorporating higher order relations. We also relied on a synthetic dataset with full 3D supervision to train our prediction networks, thereby limiting direct applicability to datasets without 3D supervision. Towards overcoming this, it might be desirable to combine our approach with parallel efforts in the community to use 2D reprojection losses (Garg et al., 2016) or leverage domain adaptation techniques (Hoffman et al., 2017).

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

## 6 APPENDIX

### 6.1 METRICS

We use the metrics from (Tulsiani et al., 2018) and summarize them below.

- Translation ($\mathbf{t}$): Euclidean distance between prediction and ground-truth $\|\mathbf{t}_\mathrm{p} - \mathbf{t}_\mathrm{gt}\|$. $\delta_\mathbf{t} \leq 0.5$m.
- Scale ($\mathbf{s}$): We measure the average unsigned difference in log-scale, *i.e.*, $\Delta(\mathbf{s}_\mathrm{p}, \mathbf{s}_\mathrm{gt}) = \frac{1}{3}\sum_{i=1}^{3} |\log_2(\mathbf{s}_\mathrm{p}^i) - \log_2(\mathbf{s}_\mathrm{gt}^i)|$. We threshold at $\delta_\mathbf{s} \leq 0.2$.
- Rotation ($\mathbf{q}$): Geodesic distance between rotations $\frac{1}{\sqrt{2}}\|\log(R_\mathrm{p}^\intercal R_\mathrm{gt})\|$. $\delta_\mathbf{q} \leq 30°$. For objects that exhibit rotational symmetry, we use the lowest error across the different possible values of $R_\mathrm{gt}$.
- Shape ($\mathbf{V}$): Following (Choy et al., 2016), we measure the intersection over union (IoU) and use threshold $\delta_V = 0.25$. As a higher IOU is better, so we use $\delta_V \geq 0.25$ for true positive.
- Bounding Box overlap ($\mathbf{b}$): The bounding box overlap is measured using IOU. $\delta_b \geq 0.5$.
- Detection: A prediction is considered a true positive when it satisfies the thresholds for each of the above components ($\delta_\mathbf{t}, \delta_\mathbf{s}, \delta_\mathbf{q}, \delta_V, \delta_\mathbf{b}$). We use Average Precision (AP) to measure the final detection performance.

### 6.2 TRAINING DETAILS

**Optimization.** We train our network in two stages. In the first stage of training we use ground truth boxes. We train for 8 epochs by using adam optimizer with a learning rate of $10^{-4}$. During the first 4 epochs of the training we train for relative and object specific predictions independently and during next 4 epochs of the training we optimize the whole model jointly by combining the relative and object specific estimates. In the next stage we consider overlapping proposals with IOU of over 0.7 with respect to ground truth boxes and the ground truth boxes as positive proposals to further make the model robust in the detection setting.

In the NYUv2 setting we start with a network trained on the SUNCG dataset and finetune the network for 16 epochs on the NYU train + val split and evaluate method on the test split.

**Rotation Prediction.** We defined $\Delta(R, d, t)$ as a measure of how inconsistent a predicted rotation $R$ is w.r.t the predicted relative direction distribution $d$ and relative translation $t$. Given a predicted rotation $R$, we would expect the predicted direction to align with the vector $\bar{R}\,\hat{t}$, where $\hat{t}$ is unit-normalized. Note that the predicted $d$ is a probability distribution over possible directions, and let $d^*$ denote the bin that aligns maximally with $\bar{R}\,\hat{t}$. We measure $\Delta(R, d, t)$ by combining measures of how likely this bin is with how well it agrees with the rotation and translation: $\Delta(R, d, t) = -\log\, p(d^*) + (1 - \cos(d^*, \bar{R}\,\hat{t}))$.

**Relative Importance.** We use lambda for unary importance to get $\mathbf{t}^*$ and $\mathbf{s}^*$ as 1. In case of rotation we use we weight for the relative predictions as $\min(5.0/n, 1)$ where $n$ represents number of neighbours of the object. In the detection setting we create set of valid objects which are allowed to influence the final predictions for other objects based upon the detection score. We consider objects with a score above 0.3 to be part of the valid set and only use them to get final predictions for other objects.

### 6.3 ARCHITECTURE ANALYSIS

### 6.4 ADDITIONAL VISUALIZATIONS AND RESULTS

We visualize the precision-recall curves in the detection setting using the SUNCG dataset in figure 6. We also visualize predictions for randomly sampled images in the setting with known bounding boxes in figure 7.

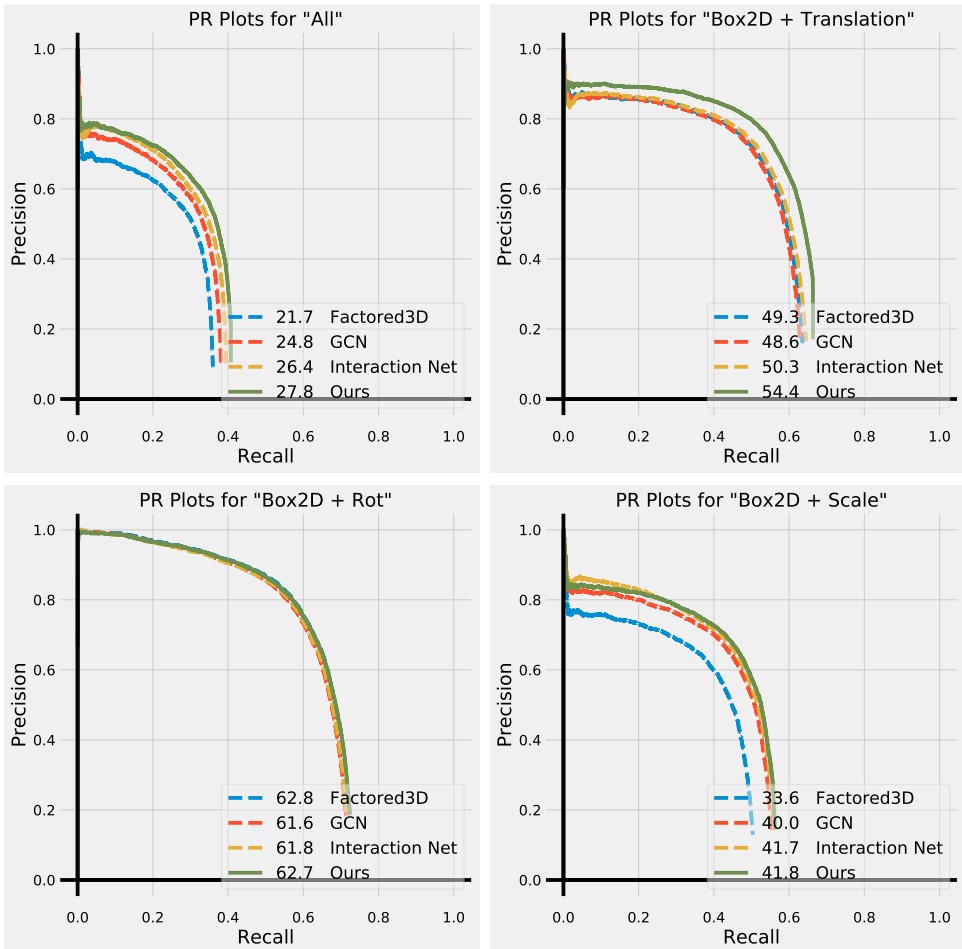

**Figure 6:** We plot the precision-recall (PR) curves for the **detection setting** for SUNCG and also display the mean Average Precision (AP) values in the legend. In each of these curves, we vary the criteria used to determine a true positive. This helps us analyze the relative contribution of each component (translation, rotation, scale) to the final performance.

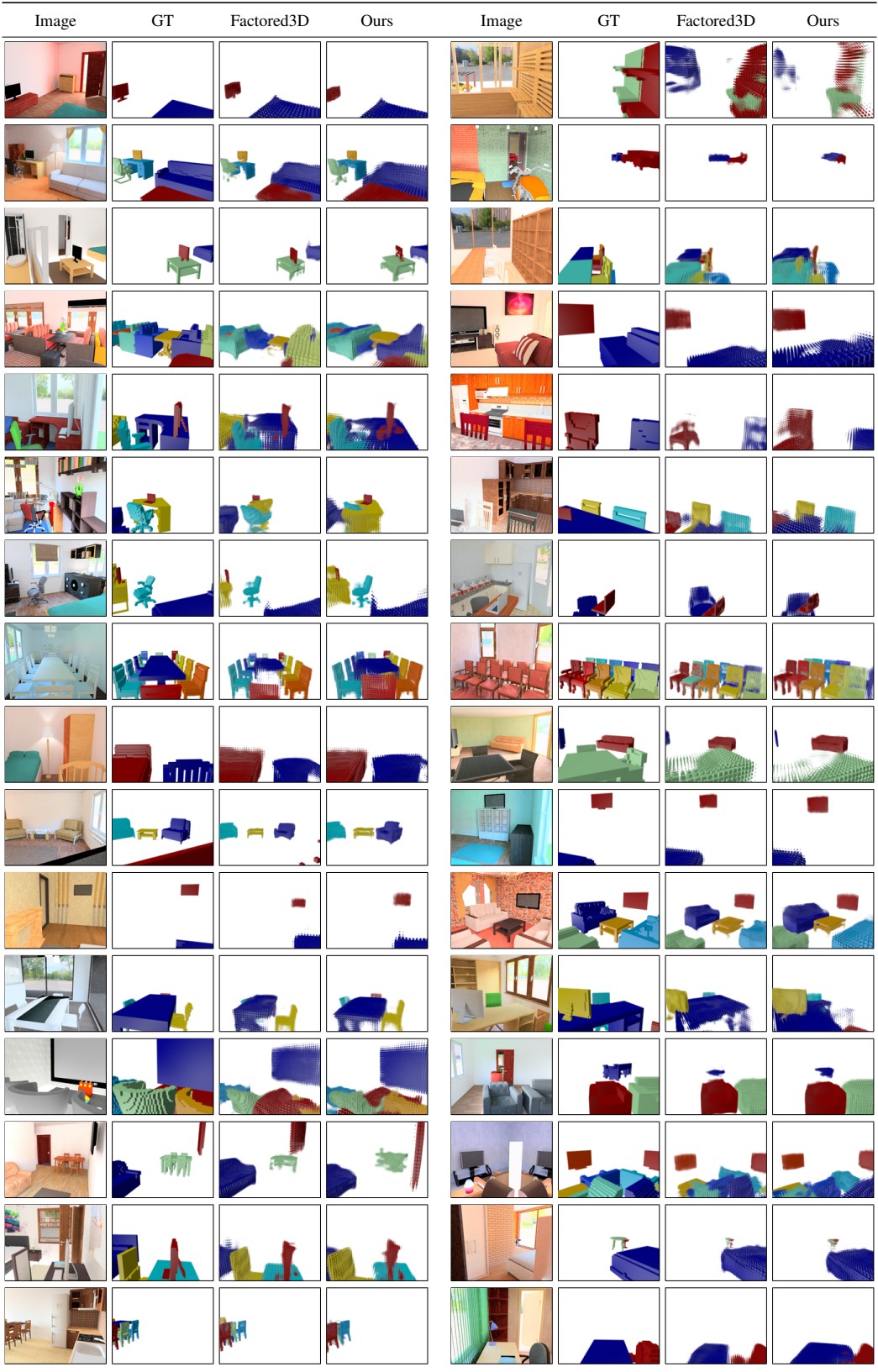

**Figure 7:** We visualize predictions for *randomly sampled images* in the setting with known ground-truth boxes for the SUNCG dataset.

