# OpenReview forum: "3D-RelNet: Joint Object and Relational Network for 3D Prediction"
_ICLR.cc/2019/Conference_

### Official Review · AnonReviewer2 · 2018-11-03
**A scene parsing model that takes both objects and their relations into account**

**Rating:** 5
**Confidence:** 5

**Review:**

This paper proposed a 3D scene parsing that takes both objects and their relations into account, extending the Factor3D model proposed by Tulsiani et al 18. Results are demonstrated on both synthetic and real datasets.

The paper is in general well written and clear. The approach is new, the results are good, the experiments are complete. However, I am still lukewarm about the paper and cannot champion it. I feel the paper interesting but not exciting, and it’s unclear what we can really learn from it.

Approach-wise, the idea of using pair-wise relationship as an inductive bias is getting popular. This paper demonstrated that it can be used for scene parsing, too, within a neural net. This is good to know, but not surprising given what have been demonstrated in the extensive literature in the computer graphics and vision community. In particular, the authors should discuss many related papers from Pat Hanrahan’s group and Song-Chun Zhu’s group (see some examples below). Apart from that, this paper doesn’t have an obvious technical innovation that can inspire future work. This is different from Factor3D, which is the first voxel-based semantic scene parsing model from a single color image, with modern neural architecture.

The results are good, but are on either synthetic data, or using ground truth bounding boxes. Requiring ground truth boxes greatly restricts the usage of these models. Would that be possible to include results under the detection setting on NYU-D or Matterport 3D? The authors claimed that the gain of 6 points is significant; however, a simple interaction net achieves a gain of 5 points, so the technical contribution of the proposed model is not too impressive.

In general, I’m on the border but leaning slightly toward rejection, because this paper is very similar to Tulsiani et al, and the proposed innovation has been explored in various forms in other papers.

A minor issue:
-	In fig 5. The object colors are not matched for GT and Factor3D and ours.

Related work
Holistic 3D Scene Parsing and Reconstruction from a Single RGB Image. ECCV’18.
Configurable 3D Scene Synthesis and 2D Image Rendering with Per-pixel Ground Truth Using Stochastic Grammars. IJCV’18.
Characterizing Structural Relationships in Scenes Using Graph Kernels. SIGGRAPH’11.
Example-based Synthesis of 3D Object Arrangements. SIGGRAPH Asia’12.

---

> ### Author Response · Authors · 2018-11-15
> **Response to  AnonReviewer2**
>
> We thank the reviewer for the comments and additional references - we will include these in related work. The reviewer raised concerns regarding the lessons learned from the paper in the context of previous pairwise relation modeling work in vision/graphics, suggested additional experiments in detection setting on real data, and pointed out a relatively small improvement over a particular baseline. Regarding the last concern, we note that apart from mAP scores we show significant improvements on errors in translations and scales as compared to all other baselines and that the difference between interaction net and our approach on these errors is significant.
>
> We hope that the discussion and results presented above in the reply to all reviewers addressed the first two concerns mentioned. We would again like to emphasize that we agree with the reviewer that previous work has examined similar inductive biases but would point out that our contributions regarding how these biases should be incorporated are novel, and would be useful for future attempts.

---

> > ### Comment · AnonReviewer2 · 2018-11-30
> > **Thanks**
> >
> > Thank you for your reply and revision. I think the revision is better than the original paper, but still, this paper is more about getting slight performance gain than proposing a novel learning method. Compared with the so-called 'implicit' methods like INs, the performance gain is relatively minor, and the idea of "using relative pose as a pairwise relation" is more like a task-specific trick during implementation.
> >
> > I'm not saying tricks are not good; if properly explained and systematically investigated, they can push the field forward. But this also means the paper would attract much more interest in conferences like CVPR/ICCV, but not ICLR, as the major ICLR audience would not be interested in contributions of such kind. I'd give a 6 if this is submitted to CVPR, but I'm keeping my 5 for ICLR.

---

### Official Review · AnonReviewer1 · 2018-11-05
**Incremental technical novelty, issues with experiments and results, unclear presentation**

**Rating:** 3
**Confidence:** 5

**Review:**

<Summary>: This paper presented a method for incorporating binary relationship between objects (relative location, rotation and scale) into single object 3d prediction. It is built on top of previously published work of [a] and used same network architecture and loss as of [a] and only added the binary relations between objects for object 3d estimation. The results are shown on SUNCG synthetic dataset and only *4 image* instances of NYUv2 dataset which is very small for a computer vision task.

[a] Shubham Tulsiani, Saurabh Gupta, David Fouhey, Alexei A Efros, and Jitendra Malik. Factoring shape, pose, and layout from the 2d image of a 3d scene. In CVPR, 2018.

<Pros>: The paper tackles a problem of obvious interest to computer vision research community. It shows better results compared to previous similar work of [a] without considering binary relation between objects.

<Cons>:

*Technical details are missing:

The set of known and unknown variables are not clear throughout the paper:
-The extrinsic camera parameters are known or estimated by the method?
-The intrinsic camera parameters are known or estimated by the method?
-What are the properties of ground truth bounding boxes in 2D camera frame and 3D space?
-What is the coordinate of translation? is it in camera coordinate or world coordinate?
-What are the variations of camera poses in training and testing for synthetic dataset and how are the samples generated? Are the train/test images generated or are rendered images from previously published work of [b] used?

[b] Yinda Zhang, Shuran Song, Ersin Yumer, Manolis Savva, Joon-Young Lee, Hailin Jin, and Thomas Funkhouser. Physically-based rendering for indoor scene understanding using convolutional neural networks. In CVPR, 2017.

*The proposed method is trained on synthetic dataset of SUNCG and their object relations have biases from scene creators. While using binary relation between objects increase the recall in prediction it can also make the predictions bias to the most dominant relations and decrease the precision of detection in rare cases in synthetic dataset. Also, such bias can decrease prediction precision in images of real scenes.

*One of the main issues in this paper is that the result of fully automated pipeline versus having ground-truth annotation at test time are mixed up. For example, in the teaser figure (Figure 1-b), does the proposed method use ground truth bounding boxes or not? It is mentioned in figure caption: “(b) Output: An example result of our method that takes as input the 2D image and generates the 3D layout.”. Is the input only 2D image or 2D image + ground truth object bounding boxes?
In order to make sure that reader understands each qualitative result, there should be a column showing the “Input” to the pipeline (Not “Image”). For example, in Figure 3 and Figure 4, the image overlaid with input ground-truth bounding boxes should be shown as input to the algorithm.


*The experiments and results does not convey the effectiveness of the proposed approach. There are major issues with the quality of the experiments and results. Here are several examples:

- Missing baseline: Comparison with the CRF-based baseline is missing. This statement is not convincing in the introduction: “One classical approach is to use graphical models such as CRFs. However, these classical approaches have usually provided little improvements over object-based approaches.” For a fair comparison with prior works, reporting results on a CRF-based baseline using similar unary predictions is necessary.

-The experimental results are heavily based on ground truth boxes for the objects, but it is not clear how/where the ground truth boxes are given at the test time and which part is actually predicted.

-If the ground truth boxes are given at the test time, it means that the ground truth binary relations between objects are given and it makes the problem trivial.

-It is not clear what is the ground truth box in experimental setup. Is it amodal object box or the ground truth box contains only the visible part of the object?

-The qualitative results shown in Figure 4 have full objects in voxel space with predicted rotation, scale and translation. In the qualitative result of Figure 3 and Figure 5 the voxel prediction is shown as final output. Why the result of full object in voxel space with predicted (rotation, scale and translation) is not shown in Figure 3 and Figure 5 and why it is shown in Figure 4?


*Very limited results on real images:

-Quantitative result on a dataset of real images is missing. The results on synthetic datasets is not a good proxy for the actual performance of the algorithm in real use cases and applications.

- The paper only shows few results of NYUv2 on known ground truth boxes. The errors in object detection can be propagated to the 3D estimation therefore these qualitative results are not representative of the actual qualitative performance of the proposed algorithm. Several randomly selected qualitative results on a dataset of real images “without ground-truth boxes” are needed for evaluating the performance of the proposed method on real images.

-Reporting variation in all parameters of scale, rotation and translation is necessary in order to find the difficulty of the problem. For example, what is the distribution of object scale in different object categories. What is the error of scale prediction of we use mean object scale for each object category for all object instance at test set?


*Unclear statements and presentation:

- It is mentioned in the paper: “While the incorporation of unary and relative predictions can be expressed via linear constraints in the case of translation and scale, a similar closed form update does not apply for rotation because of the framing as a classification task and non-linearity of the manifold.”

-Is it necessary for the relative rotation to be formulated to classification task?

-If not the comparison of modeling relative rotation via linear constraints is missing.

- In some of the tables and figures the “know ground-truth boxes/detection setting” are in bold face and in some cases are not. This should be consistent throughout the paper.

---

> ### Author Response · Authors · 2018-11-15
> **Response to AnonReviewer1**
>
> We thank the reviewer for the comments. Below we address specific concerns regarding the work.
>
> [Missing details]
> We point out relevant sections in the original submission where we already had mentioned the details.
>
> - Translation in camera vs. world frame: In Section 3.1, we outline the parameterization of the 3D pose in terms of translation, rotation, and scale in the camera frame.
> - Rendered images and pose variations: As we mentioned in Section 4.1 (Experimental Setup). We use the rendered images provided by Zhang et al.[1] Their work generated these images with a variety of different camera poses. We randomly partition this set of images into three splits from different houses - 70% (train), 10% (val) and 20% (test).
> - Extrinsic vs. Intrinsic camera parameters: We estimate the pose of objects in the camera frame we can say that the extrinsics are Identity. Also, we make no assumptions about Intrinsics of the camera. This is outlined in Section 3.1 (page 3).
>
> - Properties of ground truth boxes in 2D: In Section 4.2, we explain that we only input the ground truth boxes in 2D. We are unclear on what the reviewer means by “properties” of bounding boxes in 3D space but would be happy to clarify if the reviewer can elaborate.
>
> - What part is predicted at test time: In Section 4.2, the ground-truth 2D boxes and the image are given as input and the method predicts the 3D pose and shape of each object. In Section 4.3, the image is given as input and the method first detects the objects and then predicts the 3D pose of each object. We predict the voxel of the object for the experiments on the SUNCG dataset.
>
> - No voxel output in Figure 4: As we mention in the Figure 4 caption - "We use the ground-truth meshes for visualization due to lack of variability in shape annotations for learning." (We have also described this in section 4.1, NYUv2 setup)
>
> - "object relations have biases from scene creators" (in SUNCG): We appreciate the concerns about the biases in the synthetic data, but note that we do show results on the real images from the NYUv2 dataset.
>
> Unclear presentation: We thank the reviewer for these suggestions. They will surely help us improve the quality of the paper.
> - Set of knowns vs. Unknowns: We will clarify this in the paper. Your suggestion of using "Input" in the figures as opposed to "Image" is a great one and will be incorporated.
> - Figure 1 (b): The input to the method is the 2D image and the associated ground-truth object bounding boxes. The output is the 3D pose and shape for each object.
> - Amodal box vs. full box: We use 2D bounding boxes as is standard in the object detection literature. These boxes are *not* amodal.
> - Relative rotation as classification, and (possible) missing comparison to linear baseline: As quaternion algebra is non-commutative the quaternion "vector" space is not linear. Thus, we formulate relative rotation as a classification problem, and hence there is no linear baseline. Also, note that previous works have also modeled rotation as a classification problem.
>
> Incorrect statements by the reviewer
> - Ground truth boxes are given implies relations are given: This is incorrect. The boxes are in 2D while the relations we use and predict in the paper are in 3D coordinates.
> - "Quantitative result on a dataset of real images is missing": Please see Table 1 that shows quantitative results on the NYUv2 dataset.
> - "*4 image* instances of NYUv2 dataset": This statement is a mis-characterization of our work. We only visualize results of a few images but note that Table 1 reports evaluations using *all* (654)  test set images on NYUv2. This is explained in Section 4.1.
>
> Missing baseline
> - CRF baseline: While this is an excellent suggestion, there are many practical reasons why it is not straightforward for us to do this. CRF based methods typically rely on class-specific pairwise potentials, and while these can be designed for specific cases (e.g. chair-table), there are numerous non-trivial design decisions that make this not scalable for generic classes e.g. what is the form of potential function, do all (or only nearby) object pairs have an edge between them, do all classes pairs have potential functions etc. However, if the reviewer has any specific suggestions regarding a CRF baseline that can generically handle 3D pose across classes, we would be happy to compare. We would also like to point out the GCN baseline, which also does message passing, can be considered as an implicitly learned CRF.

---

### Official Review · AnonReviewer3 · 2018-11-06
**The authors propose an end-to-end trainable model which leveraged pair-wise relationships between objects to predict objects 3D shape and pose given a single 2D image.  The proposed method outperforms independent prediction approaches on two publicly available datasets.**

**Rating:** 6
**Confidence:** 4

**Review:**

The paper is well-written with a few figures to illustrate the ideas and components of the proposed method. However, one of the main components in the proposed method is based on Tulsiani et al. CVPR'18. The remaining components of the proposed method are not very new. Hence, I am not very sure whether the novelty of the paper is significant. Nevertheless, the performance of the proposed method is fairly good outperforming all baseline methods.
I also have a few questions:
1. How did you get the instance boxes, union boxes, and binary masks in testing?
2. What are the training and inference time?

---

> ### Author Response · Authors · 2018-11-15
> **Response to AnonReviewer3**
>
> We are grateful for your feedback. We hope that the above discussion assuaged the reviewer’s concerns regarding novelty and some unclear details. We briefly address the two questions regarding the setup:
>
> During testing, in the setting with known GT boxes (Sec 4.2), we assume that the 2D instance boxes are given. In the detection setting, the 2D instance boxes are the result of the learned detector. Given the (detected or known) instance boxes, the union boxes and binary masks can be easily computed - the union box is just the larger box containing both instance boxes, and the mask highlights these instance boxes in the union box.
> Training and Testing  Inference Time on a single GPU (Maxwell Titan X)
> 1. Train time: 65 hrs
> 2. Test time: 0.55s per image

---

### Author Response · Authors · 2018-11-15
**Overall Response and Clarifications**

We would like to thank the reviewers for their helpful comments and valuable feedback. We first address the common questions raised by the reviewers and then talk about reviewer specific queries. Major changes to the pdf post reviews are colored in *blue* to enhance readability. While AR1 gives this work the lowest rating among reviewers, we would like to highlight that some of the raised concerns/issues are incorrect (for eg, no quantitative results on real images -- we have performance number on NYUv2 which is a real-world indoor image dataset), and we hope that they would reconsider their rating in this light.

A common concern among reviewers seems to be - “Is our work significant enough?”
We believe it is; in terms of both -- empirical results and technical novelty.

[Empirical Results]: In computer vision, significant advances have been made by making progress on benchmarks. Our paper represents a simple yet effective approach which advances state of the art significantly on NYUv2 (and SUNCG as well). We believe the computer vision community deserves to know the new benchmark performance so it can build upon this.

[Technical Novelty]: As AR2 points out, our work like several others in 3D scene modeling and other areas leverages the use of pairwise relations as an inductive bias, and the reviewers feel our contribution is not significant in this light. We would first like to emphasize that the questions of whether an inductive bias is useful, and how its inclusion is operationalized are both extremely important, and research along the latter is no less useful despite evidence for the former -- object detection methods have over the years refined how object proposals can be used, investigating how geometry can be used as an inductive bias for learning based 3D reconstruction is a popular recent area, etc.

In addition to arguing for the need of modeling relationships (which we agree has been previously explored), our contribution in this work also relates to investigating how this inductive bias of leveraging pairwise relationships should be incorporated, and we propose an approach different from previous implicit (e.g. IN, RN) or prior-based methods (e.g. existing work in vision/graphics).


The previous vision/graphics approaches rely on image-agnostic category-level priors, we instead propose to use image-dependent pairwise prediction.


Unlike implicit methods, our ‘relations’ pertain to specific variables of interest i.e. relative pose and are combined during learning with the unary estimates.


We note that both these modeling insights: i) image-dependent pairwise prediction, and ii) using relative pose as a pairwise relation, are contributions that future work can draw upon, and we do not feel that these proposed innovations have been previously explored.

[Additional Experiments on Real Data]:
AR1 incorrectly comments that “Quantitative result on a dataset of real images is missing” - we do report quantitative results on the NYU dataset (see Table 1) in the setting with known GT boxes. AR3 additionally suggested evaluating our approach in detection setting on a real dataset to further strengthen the empirical validation of the approach. We did so on the NYUv2 dataset, and obtained an AP of 8.5 for our method (compared to Factored3D: 5.3, GCN: 6.5, InteractionNet: 7.6). The lower absolute numbers indicate that the difficulty of the full task, but we note that the relative performance trends are similar to SUNCG (in fact, our relative improvements over baselines are larger). We have included the full table and more details in the paper.

[Details on Setup]:
AR1 and AR3 found some aspects of the setup unclear. We emphasize that the experimental setup is exactly the same as followed in the previous work we compare to (Factored3D). The task of predicting 3D for all objects are considered in two evaluation settings: -
[Sec 4.2] Known GT 2D bounding box - the input is an RGB image and the 2D bounding boxes for all objects are input, and we estimate the 3D shape and pose for all given objects.
[Sec 4.3] Detection setting - only the 2D image is input, and we perform detection as well as 3D estimation for all detected objects (the bounding boxes come from a pretrained object-detector).

Note that in addition to the above input, we do not use any other annotation at inference. We clarify the other specific questions is the individual responses to reviewers and will make the setup clearer in the main text.

---

> ### Comment · AnonReviewer1 · 2018-12-04
> **Still unclarified issues and major shortcomings in the experiments. Novelty is incremental.**
>
> I read authors’ rebuttal. In the rebuttal some of my questions and concerns are quoted incompletely such that my original statements can be interpreted in a different way than what I had originally written. Then, the authors have used such incomplete quotations and have called them as “incorrect statements of the reviewer”. I do not appreciate such practice of *faulty quotation* of reviewers' statement as *incorrect statement*. Such practice can also confuse AC, other reviewers and all other paper audiences.
>
> After reading rebuttal, I want to keep my original score of “3: Clear rejection” for this paper as I still see major shortcomings with the experiments and results and I am not convinced that this work has sufficient novelty for publication. Also, there are still unclarified issues with the paper.
>
>
> Here are my post-rebuttal comments:
>
> Based on the authors’ response in rebuttal, authors use identity matrix as extrinsic camera parameter and do not make any assumption about intrinsic parameter of the camera. This means that the proposed method learns the statistics of the object appeared in the synthetic scenes in the camera frame. This is *not* 3D estimation of the objects in the scene, it is called 2.5D estimation in computer vision literature. Since the proposed method learns the model on synthetic rendered images of [Zhang et al., 2017], it learns the 2.5D estimation of objects in the camera field of view on that dataset. This causes degradation of performance and reduces applicability of the learned model in real use cases.
>
> There is no qualitative result on real dataset of NYUv2 in *detection setting*. The qualitative result in *ground-truth bounding box setting* is far from real use case and application. The table for numerical comparison in detection setting is added to the revision, however, meanAP degradation is a lot from synthetic dataset to real dataset (27.76 in synthetic dataset of SUNCG to 8.49 in real dataset of NYU).
>
>
> I requested reporting of the statistics of all variables including scale, rotation and translation in the data that is used in the paper for understanding the difficulty of the problem. Authors did not provide any response to this request.
> For example, what is the distribution of object scale in different object categories. What is the error of scale prediction when you use “average scale” for all objects instances of the same category in the test set?
>
> I also requested baseline comparison with a CRF-based method, but no CRF baseline is conducted in revision and authors’ response about “numerous non-trivial design decisions” in CRF-based methods in the literature is not convincing.
>
>
> -Authors did not give proper credit to [Zhang et al., 2017] for the dataset of Physically-based rendering of indoor scene images:
>
> [Zhang et al., 2017] Yinda Zhang, Shuran Song, Ersin Yumer, Manolis Savva, Joon-Young Lee, Hailin Jin, and Thomas Funkhouser. Physically-based rendering for indoor scene understanding using convolutional neural networks. In CVPR, 2017.
>
>
> One of the main contributions of [Zhang et al., 2017] is "Physically-based rendering for indoor scene" and the credit for the realistic rendering of synthetic dataset of RGB images should be given to [Zhang et al., 2017], not [Zhang et al., 2017; Tulsiani et al., 2018].
> Currently the statement in the manuscript in Sec 4.1 page 5 is:
> “Following (Zhang et al., 2017; Tulsiani et al., 2018), we use the 2D renderings of the houses and the corresponding parsed 3D house model information to get roughly 550k image and 3D supervision pairs.”

---

### Meta-Review · Area_Chair1 · 2018-12-13
**Reviews not strong enough to justify acceptance**

**Confidence:** 4
**Recommendation:** Reject

**Metareview:**

With ratings of 6, 5 & 3 the numerical scores are just not strong enough to warrant acceptance.
The author rebuttal was not able to sway opinions.